# A miniature dialysis-culture device allows high-density human-induced pluripotent stem cells expansion from growth factor accumulation

Fuad Gandhi Torizal [1,2✉], Qiao You Lau [1], Masato Ibuki[3], Yoshikazu Kawai[3], Masato Horikawa[4], Masataka Minami[4], Tatsuo Michiue [5], Ikki Horiguchi[6], Masaki Nishikawa[2] & Yasuyuki Sakai[1,2]

Three-dimensional aggregate-suspension culture is a potential biomanufacturing method to produce a large number of human induced pluripotent stem cells (hiPSCs); however, the use of expensive growth factors and method-induced mechanical stress potentially result in inefficient production costs and difficulties in preserving pluripotency, respectively. Here, we developed a simple, miniaturized, dual-compartment dialysis-culture device based on a conventional membrane-culture insert with deep well plates. The device improved cell expansion up to approximately ~3.2 to $4 \times 10^7$ cells/mL. The high-density expansion was supported by reduction of excessive shear stress and agglomeration mediated by the addition of the functional polymer FP003. The results revealed accumulation of several growth factors, including fibroblast growth factor 2 and insulin, along with endogenous Nodal, which acts as a substitute for depleted transforming growth factor-β1 in maintaining pluripotency. Because we used the same growth-factor formulation per volume in the upper culture compartment, the cost reduced in inverse proportional manner with the cell density. We showed that growth-factor-accumulation dynamics in a low-shear-stress environment successfully improved hiPSC proliferation, pluripotency, and differentiation potential. This miniaturised dialysis-culture system demonstrated the feasibility of cost-effective mass production of hiPSCs in high-density culture.

[1] Department of Bioengineering, Graduate School of Engineering, The University of Tokyo, Tokyo, Japan. [2] Department of Chemical Systems Engineering, Graduate School of Engineering, The University of Tokyo, Tokyo, Japan. [3] Regenerative Medicine and Cell Therapy Laboratories, Kaneka Corporation, Kobe, Japan. [4] Materials Research Laboratories, Nissan Chemical Corporation, Saitama, Japan. [5] Department of Life Sciences, Graduate School of Arts and Sciences, The University of Tokyo, Tokyo, Japan. [6] Department of Biotechnology, Graduate School of Engineering, Osaka University, Osaka, Japan. ✉email: t_gandhi@chemsys.t.u-tokyo.ac.jp

Human-induced pluripotent stem cells (hiPSCs) are a potential source to generate many cell types comprising the human body. A large number of hiPSCs are essential for translational applications, such as regenerative therapy. At present, suspension culture is a common standard for producing large numbers of pluripotent stem cells (PSCs)[1,2]; however, large-scale and cost-effective hiPSC production remains difficult because of several culture-condition requirements, including nutrition-supply limitations, waste-product removal[3], and damage by mechanical stress[1]. In addition, mass production of hiPSCs requires a large number of expensive growth factors. These problems can potentially lead to high production costs[4].

Dialysis-culture systems were first utilized to produce various proteins, such as monoclonal antibodies[5–7] and tissue-type plasminogen activator[8], from mouse hybridoma cell lines cultured at high density. Dialysis membranes permeate small molecules, such as toxic metabolites and nutrition sources, but not macromolecules, such as proteins secreted from cells. Several studies have applied this technology for PSC culture using large culture vessels at low cell density[9,10]; however, the feasibility of high-density PSC culture has not been examined in-depth owing to the cost and complexity of operations in these large-scale systems. Furthermore, PSCs secrete several autocrine factors necessary to improve their proliferative efficiency[11–13], and these factors can be fully utilized and evaluated using this dialysis system[14].

Hydrodynamic conditions are important factors for controlling the aggregate size and avoiding excess agglomeration, which can cause spontaneous differentiation and/or necrotic cores owing to the mass-transfer limitations of oxygen, nutrition, and metabolic waste[15]. Our previous study showed the potential of dialysis culture in low-density PSC differentiation[16]; however, the performance of high-density suspension culture can result in mechanical stress imposed by both the rotation of the culture medium and collision between aggregates. A biopolymer comprising gellan gum is a biocompatible material used for various tissue-engineering purposes[17,18]. Otsuji et al.[19] and Horiguchi et al.[20] showing that the addition of this biopolymer can potentially support the suspension culture of PSCs by reducing excess agglomeration and cell damage caused by shear stress.

In this study, we assessed the feasibility of high-density expansion of hiPSCs in a simple, miniaturized, dialysis-culture system capable of evaluating the effects of various medium components, including nutrition, growth factors, and toxic metabolites, on cell proliferation, pluripotency, and differentiation capacities without the need to use a large amount of culture medium and supplemental growth factors. Because high-density culture can increase mechanical stress not only caused by shear stress but also aggregate collision[21], we applied FP003 functional polymer solution containing gellan gum to create a low-mechanical-stress culture environment by increasing medium viscoelasticity. We found that endogenous autocrine factors accumulated in this system and that their effective use minimized the demand for their exogenous supplementation. The results demonstrate that the expansion of high-density hiPSCs was enabled by both the accumulation of endogenous autocrine factors and the low-mechanical-stress environment from using the dialysis membranes and FP003. Moreover, the increased cell density not only reduced the use of growth factors but also supported the pluripotency and differentiation potential of hiPSCs. Although this study was limited to the proliferation phase, the possibility of reducing costs by full utilization of both endogenous and exogenous growth factors at high cell density provides an insight into large-scale organ-cell production from human stem cells and could be effective at differentiating to specific cell lineages that require more complex and expensive growth-factor-based protocols.

## Results

**Mass-transfer of the dialysis culture system under cell-free conditions.** The dialysis-culture system was performed to continuously support medium refinement through the dialysis membrane (Fig. 1a). The permeability evaluation shows that this simple dialysis device was able to maintain the equilibrium of glucose concentration after 6 h of culture and attain equilibrium of the lactic acid concentration in 8 h (Fig. 1b, c).

We confirmed the ability of the device to retain high-molecular weight growth factors using a permeability test with fluorescein isothiocyanate (FITC), which evaluates the capability of size-selective accumulation of macromolecules, such as growth factors. During 24 h, the only macromolecule capable of passing through the 12-kDa molecular weight cutoff (MWCO) dialysis membrane was a very small amount of 4-kDa FITC, whereas 10-kDa FITC and 20-kDa FITC remained in the upper culture compartment with no detection in the lower dialysate compartment (Fig. 1d–f).

To achieve a mild dynamic culture environment, we added FP003 solution to the medium in the upper culture compartment. This biomaterial reduces shear stress and prevents excess agglomeration between aggregates in high density by forming microfiber linked by calcium ions (Fig. 1g). The findings show that FP003 addition in the culture medium did not alter the mass-transfer capability between the two compartments (Fig. 1b, c).

**The effects of FP003 on shear-stress-induced apoptosis or necrosis.** To evaluate possible secondary apoptosis or necrosis caused by the dynamic culture conditions, the hiPSC aggregates were cultured for 24 h under rotational culture in the presence and absence of FP003. The results showed that the addition of FP003 can reduce cell death. This condition was confirmed by the measurement of lactate dehydrogenase (LDH) leakage caused by membrane rupture of damaged cells as an indicator of secondary apoptotic or necrotic cells (Fig. 1h), and Caspase 3/7 activity as an indicator of primary apoptotic cell death (Fig. 1i). These findings also show a higher number of secondary apoptotic or necrotic cells rather than primary apoptosis, which means that the mechanical stress-induced necrosis was a prominent cause of cell death during the high-density culture, and it was ameliorated by FP003 in the culture medium.

**Improved cell proliferation and aggregate morphology.** When the single-cell suspension was directly inoculated in the dialysis culture compartment, excessive aggregation was observed. Therefore, the aggregates were thus pre-formed using six-well plates at a rotation of 90 rpm before the dialysis culture. After the aggregates were formed, they were transferred to a smaller dialysis culture insert and rotated at 120 rpm for the uniform dispersion of the aggregates. These rotational speeds were mainly decided by the outward centrifugal force and the inward force of the swirling flow and they were experimentally optimized for each scale.

After expansion in different culture configurations (Fig. 2a), the hiPSC aggregates showed different morphologies and proliferation levels. The aggregate population expanded in simple dialysis culture (D/MR(−)) showed decent morphology (Fig. 2b) with a tolerable size for expansion, as indicated by the absence of necrotic areas at the center of the aggregate, indicated by histological analysis (Fig. 2c). This necrosis was often caused by the mass-transfer limitation of oxygen, nutrients, and waste products in large aggregates. The aggregates population expanded in this system also showed the increasing diameter (Fig. 2d), resulting in an increase of cumulative cell proliferation during five passages (Fig. 2e) within a high-density cell number (Fig. 2f, g). By contrast, the proliferation rate of all hiPSCs line populations in a conventional culture system with manual daily medium replacement (C/MR(+)) was limited to ~$5 \times 10^6$ cells/mL at the

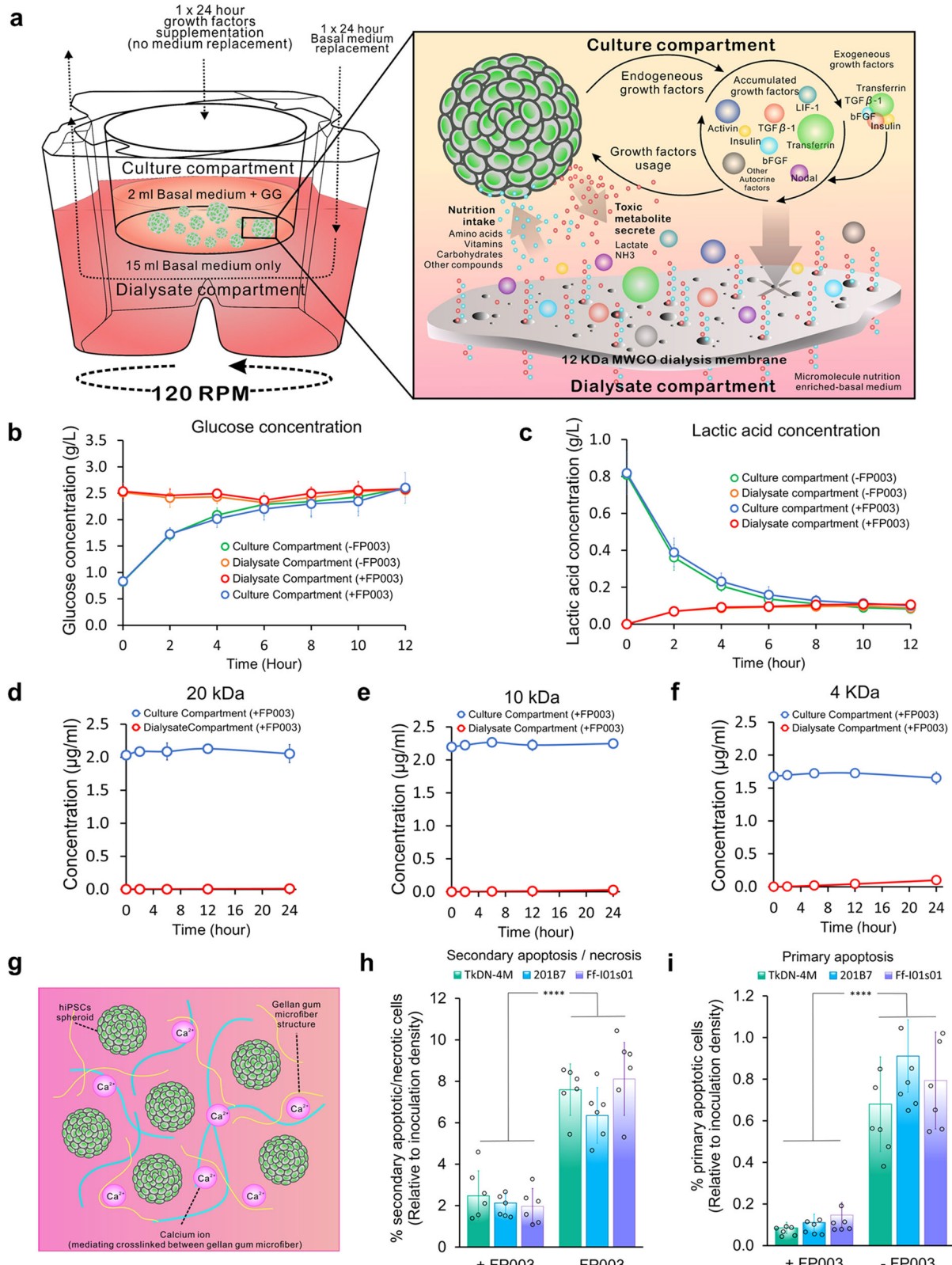

end of the expansion period, and the extreme cell number depletion occurring during the first and second days of culture without any medium replacement (C/MR(−)) (Fig. 2f, g).

**Continuous nutrient supply and toxic metabolite removal.** Based on glucose measurements, the dialysis systems demonstrated

higher performance in maintaining glucose levels as compared with the other two groups. According to the lower dialysate compartment, glucose concentrations in the dialysis systems were maintained at between ~1.8 g/L and 3.0 g/L, which was nearly the original concentration of the culture medium (Fig. 3a). These results showed that in terms of glucose delivery, the dialysis support in D/MR(+) was capable of supporting

**Fig. 1 The main concept, mechanism, and evaluation of simple dialysis culture with FP003 addition. a** The schematic concept of high-density hiPSCs expansion in a miniaturized dual-compartment dialysis system. The permeability of small molecules such as **b** glucose and **c** lactic acid was also confirmed during 12 h ($n = 3$ biologically independent experiments). There is no difference in the mass transfer of these molecules between the normal medium without FP003 (−FP003) and with FP003 (+FP003). The cell-free penetration assays of various large molecules: **d** 20 kDa FITC, **e** 10 kDa FITC, **f** 4 kDa FITC show a decent capability to accumulate the macromolecule ($n = 3$ biologically independent experiments). **g** The mechanism of the FP003 addition in the culture medium to prevent excess agglomeration and mechanical stress. **h** The mechanical stress-induced secondary apoptosis or necrosis measured by intracellular lactate dehydrogenase (LDH) release ($n = 6$ biologically independent experiments) and **i** Primary apoptosis was reduced during 24 h of culture in gellan gum addition ($n = 6$ biologically independent experiments). To obtain the approximate number of secondary apoptotic/necrotic and primary apoptotic cells, the resulting value of LDH or caspase 3/7 assays was calibrated to the value of the graded number of intentionally necrotic- or apoptotic induced hPSCs line. The percentage shows the approximate number of secondary apoptotic/necrotic and primary apoptotic cells from the total inoculation cell number. Mean ± standard deviation are indicated in each graph. Statistical significance: ****$p < 0.0001$.

feeding requirements of small molecule nutrition for high-density culture.

The lactate concentration in the dialysis-culture system was successfully maintained at a lower range than critical lactate concentration based on previous study[23,24], which can lead to cell damage and decreased pluripotency (Fig. 3b). In addition, we confirmed this by higher lactate accumulation in groups without dialysis and with and without 24-h medium replacement, in which lactate concentrations rose above the critical limit.

Interestingly, despite the increasing level of lactate concentration and depletion in cell viability, the glucose level was remained detected in the group without any medium replacement in C/MR(−). This showed that instead of glucose starvation, lactate accumulation was the main limiting factor for cell proliferation. This phenomenon was further confirmed by the critical lactate concentration in conventional daily medium replacement in C/MR(+), where the cell growth was limited up to ~$5 \times 10^6$ cells/mL. As such, the daily medium replacement was not enough to handle the lactate accumulation in the small 2 mL volume.

**Growth-factor accumulation in culture medium.** Without the necessity for complete growth factors containing medium replacement in the dialysis-culture system, we observed accumulation of exogenous growth factors, such as fibroblast growth factor 2 (FGF2), during the expansion period. By contrast, conventional daily replacement of the medium completely removed the remaining FGF2 (Fig. 4a). A similar phenomenon was observed with insulin accumulation, which accumulated in the dialysis culture and promoted continuous hiPSC proliferation relative to the other culture systems (Fig. 4b). However, we found that transforming growth factor-β1 accumulation depleted overtime in the dialysis culture (Fig. 4c).

In addition, to evaluate whether if there is any interference of endogenous autocrine factors produced by the hiPSCs, we choose Nodal. According to the medium formulation in this study (Table 1), we did not exogenously supplement Nodal, which was confirmed by its absence in the culture medium at day 0. This protein is a member of the TGF-β family and is natively secreted by PSCs[25]. Together with other TGF-β family members, such as TGF-β1 and activin A, a sufficient amount of Nodal is essential to regulate pluripotency[25,26]. The Nodal accumulation was confirmed by its coexistence in the culture medium. Because Nodal acts within the similar TGF-β pathway, it might represent an autocrine backup in maintaining pluripotency, suggesting that Nodal accumulation can contribute to the total cost reduction of the dialysis system by reconstituting the need for further TGF-β1 addition. These results showed that at the end of the expansion, Nodal concentration increased over time in culture medium from dialysis culture based on its accumulation, better than its levels in conventional culture systems both with and without medium replacement (Fig. 4d).

**High-density culture improves pluripotency and differentiation potential.** Characterization of hiPSCs after expansion showed that the aggregates growing in the dialysis culture exhibited higher levels of various pluripotency markers. Alkaline phosphatase activity assay showed higher metabolic activity in the aggregates expanded in dialysis culture than those in the other cultures (Fig. 5a). This result corresponded with their higher protein level analysed by immunocytochemistry (Fig. 5b). The pluripotency state was indicated by higher stage-specific embryonic antigen 4 (SSEA-4)-positive cells in all of the hiPSC lines grew in dialysis culture (D/MR(−)) compared with the one which expanded in conventional complete medium replacement culture (C/MR(+)) (Fig. 5c). In general, most pluripotency markers, such as octamer-binding transcription factor 4 (OCT4), SRY-box transcription factor 2 (SOX2), and homeobox protein NANOG, were upregulated in dialysis culture as compared with conventional suspension culture with medium replacement. Among the selected genes, expression levels of NANOG were significantly higher in all of the hiPSCs expanded in the dialysis culture system (Fig. 5d). The results of immunocytochemistry confirmed the differences in these markers at the protein level between the groups cultured in D/MR(−) and C/MR(+) (Fig. 5e).

Genomic stability after long-term culture was evaluated in all of the hiPSC lines expanded in the dialysis-suspension culture (D/MR(−)) and conventional suspension culture with daily medium replacement (C/MR(+)). The results revealed that the chromosomal abnormality occurred in the two hiPSC lines, TkDN-4M (chromosome number 17) and Ff-I01s01 (chromosome number 12) after being cultured in conventional suspension culture with daily medium replacement (C/MR(+)). In contrast, all of the hiPSCs expanded in the dialysis culture system (D/MR(−)) were able to maintain their normal karyotype until the end of the expansion (Fig. 6a).

In addition, the hiPSC aggregates that previously grew in dialysis culture showed higher differentiation capabilities into three germ layers: ectoderm, mesoderm, and endoderm, by direct differentiation assay. This population exhibited better gene-expression levels of lineage-specific markers for ectoderm (NESTIN, orthodenticle homeobox 2 (OTX2) and paired box protein 6 (PAX6)) (Fig. 6b), mesoderm [T-box transcription factor T (Brachyury), runt-related transcription factor 1 (RUNX1) and T-Box Transcription Factor 6 (TBX6)) (Fig. 6c), and endoderm (SRY-Box Transcription Factor 17 (SOX17), Forkhead Box A2 (FOXA2), and GATA-binding protein 4 (GATA4)) (Fig. 6d).

**Discussion**
One of the main problems in the clinical application of PSCs is the high cost associated with their production. Among all of the culture medium components, recombinant growth factors represent the most expensive aspect of their formulation for PSC culture. To save growth-factor usage, strategies have been employed to retain the accumulated growth factors by using

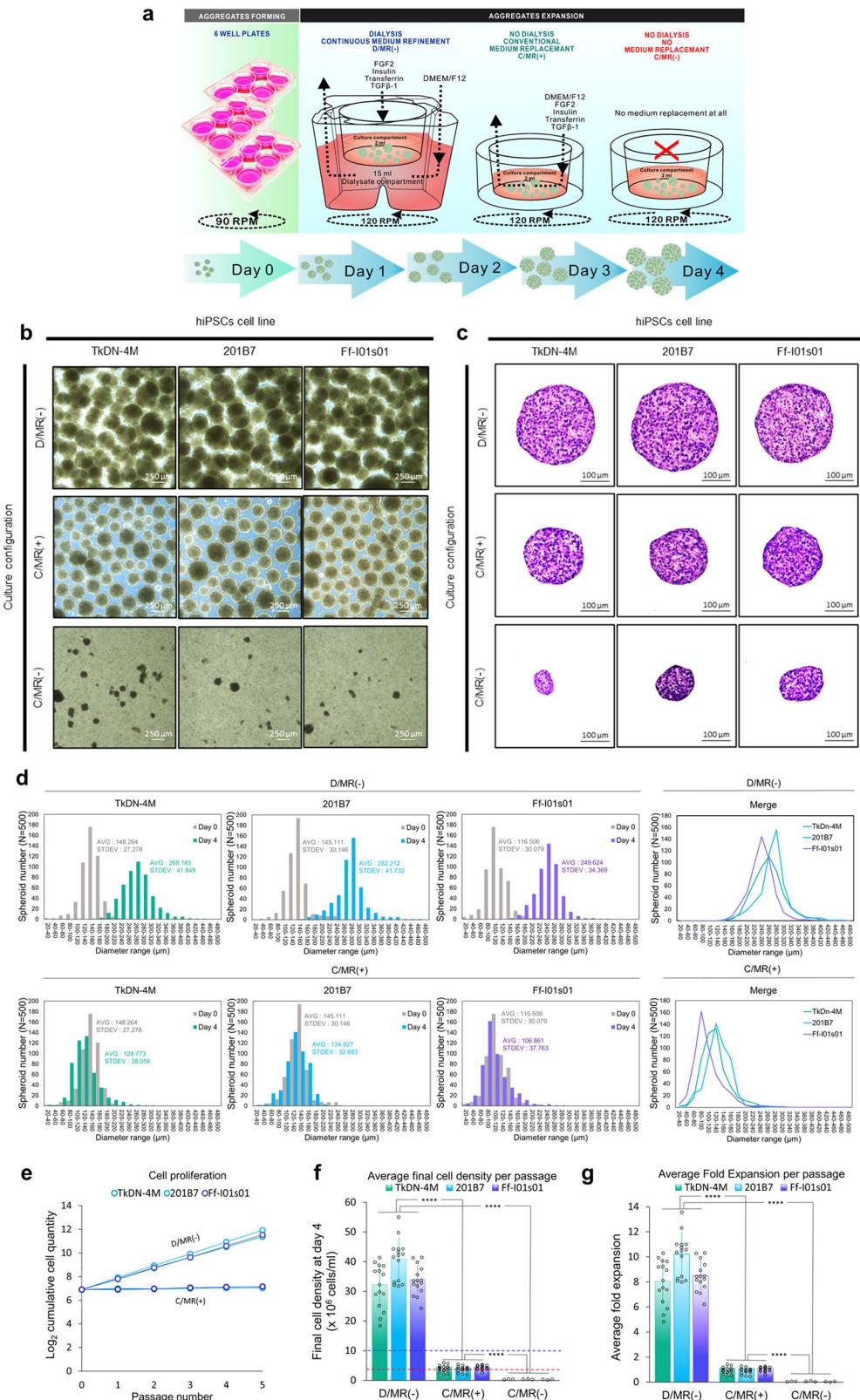

dialysis-based culture systems. However, most of these large culture systems are not well optimized to push their potential to enable low-cost expansion by highly increasing cell density (i.e., $\geq 1 \times 10^7$ cells/mL)[22]. High-density culture might also increase additional risks, such as high mechanical stress due to aggregate collision[21] and excess agglomeration[19]. To address this problem, we designed a miniaturized dialysis-culture system for the first-

line evaluation of high-density culture. Our findings showed that this system was able to support the production of hiPSCs in high-density volumetric yield up to $\sim 32-40 \times 10^6$ cells/mL with decent pluripotency preservation and differentiation capability. This high-density culture was achieved by inoculating $4 \times 10^6$ hiPSCs/mL, which showed optimum proliferation with acceptable lactate concentrations and better growth factors accumulations. The

**Fig. 2 Experimental procedure, aggregate morphology, and proliferation of hiPSCs expansion in the simple miniaturized dialysis culture system.**
**a** The illustration of daily medium change. The aggregates formed by culturing the single cell in six-well plates in 24 h before being moved into each culture system for 4 days expansion. The cycle was repeated for five-time passages and divided into three different groups: the simple dialysis-suspension culture without complete medium replacement (D/MR(−)), the conventional suspension culture with complete medium replacement (C/MR(+)), and conventional suspension culture without complete medium replacement (C/MR(−)). **b** The aggregates morphology among different culture configurations and inoculation density. **c** The absence of necrotic core was evaluated by cross-sectioned-histological analysis of resulting hiPSCs aggregates. **d** The aggregates growth was evaluated by comparing the diameter between aggregates population expanded in D/MR(−) and C/MR(+) ($N = 500$ aggregates per group). The diameter of aggregates population in C/MR(−) was excluded due to low survival after the first passage. The Dialysis-culture system was able to support the cell proliferation at high density, showed by **e** The cumulative cell quantity during five-time passages. **f** Average final cell density per passage; and **g** average fold expansion relative to inoculation cell number. The red line indicated inoculation density at day 0. The minimum criteria of high-density mammalian cell culture referring to Griffiths et al.[22] is indicated by the blue line. Mean ± standard deviation are indicated in each graph.

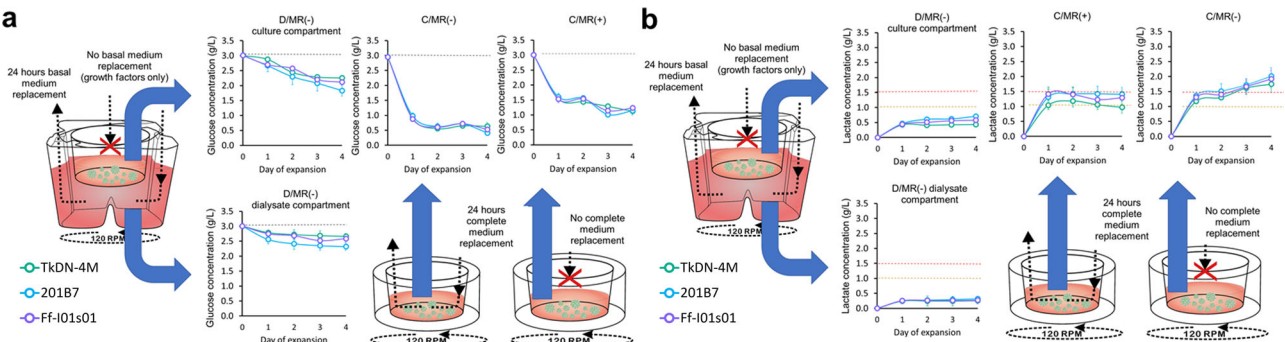

**Fig. 3 The concentration of glucose and lactate in the culture medium partially reflects the culture environment condition in a different configuration.**
**a** The glucose concentration and **b** the lactate concentration in culture medium. Gray line showed the glucose concentration in the culture medium. The critical lactate concentration for PSCs culture described by Ouyang et al.[24] indicated in red line and critical lactate concentration for PSCs culture, which described by Horiguchi et al.[23] indicated by the yellow line. (D/MR(−) and C/MR(+): $n = 3$ biologically independent experiments during five passages; C/MR(−): $n = 3$ biologically independent experiments during one passage). Mean ± standard deviation are indicated in each graph.

hiPSCs cultured in this condition also retained higher pluripotency (Supplementary Figs. 1–4). In addition, adequate hiPSCs expansion was acquired by reductions in mechanical stress and excess agglomeration via the addition of FP003 biopolymer solution. The increase in final cell density might allow a reduced requirement for expensive growth factors[9,10]. The dialysis culture mimicked in vivo conditions, where the production of autocrine factors is utilized in compact, three-dimensional tissue receiving a continuous supply of nutrition from the blood. Mimicking this phenomenon in an in vitro system allows the acquisition of healthy and high-quality PSCs in a cost-effective manner through maintenance of their autonomous homeostasis in high-density cell culture. This culture system offers insight into the utilization of exogenous and endogenous growth-factor accumulation in a cost-effective dialysis-culture system.

We observed the successful accumulation of several exogenous growth factors in the dialysis-culture medium to support further proliferation. This study was utilized the simple medium formulation developed by Chen et al.[27] based on the essential minimum components required for PSC culture, including Dulbecco's modified Eagle medium (DMEM)/F12 basal medium and supplementation with insulin, transferrin, TGF-β1, and FGF2 (Table 1)[27]. Similar to that used for other PSC cultures, this medium includes a large amount of FGF2[28] and insulin[29], whereas TGF-β1 is often supplemented in small amounts to support the pluripotency[30]. Some studies reported that some autocrine factors, such as FGF2[31–33], TGF-β1[34–36], and insulin-like growth factors[37–39], are secreted by stem cells to support their homeostasis, which is reinforced by the addition of exogenous FGF2 and insulin to the culture medium. However, the remaining excess growth factors, such as insulin and FGF2, are usually removed when the medium is replaced. Therefore, utilization of accumulated insulin and FGF2 by dialysis culture might further reduce the

costs. In addition, TGF-β1 is depleted over time in dialysis culture, making it insufficient to support the high-density hiPSCs culture. Although recent studies show that PSCs can still proliferate and maintain their pluripotency in the absence of TGF-β1 and FGF2[40,41], PSCs cultured in their absence are less dependent on glycolytic pathways, which reduces their proliferation[40]. By contrast, in the present study, we showed that gene expression associated with pluripotency remained enhanced, suggesting that support might come from other autonomous regulatory pathways.

The endogenous growth factors synthesized and secreted by three-dimensional hiPSCs aggregates contribute not only to a suitable culture environment for cell growth but also potentially supported pluripotency maintenance and differentiation capacity of hiPSCs[11]. Other reports show that better cellular metabolism and yield can be achieved by increasing inoculation density[42]. Increases in cell density might also play an important role in maintaining pluripotency through the improved accumulation of endogenous growth factors and their autocrine derivatives. To confirm their advantageous accumulation, we chose Nodal as a selected candidate factor to represent the accumulation of important hiPSC autocrine factors. Nodal and TGF-β1 levels are mediated by a similar mechanism via SMAD2/3 signaling to maintain PSC pluripotency by enhancing NANOG upregulation to maintain balanced differentiation into ectoderm or endoderm lineage[26,43–47] (Supplementary Figure 5). In response to these mechanisms, we found that Nodal levels increased in a density-dependent manner and consistently correlated with NANOG gene-expression patterns (Supplementary Figs. 3–4). This indicated that Nodal might reconstitute the role of TGF-β1 in maintaining pluripotency and retaining differentiation potential into three human embryonic germ layers cell types.

As expected, medium refinement by the dialysis-culture system successfully supported further hiPSC proliferation. To

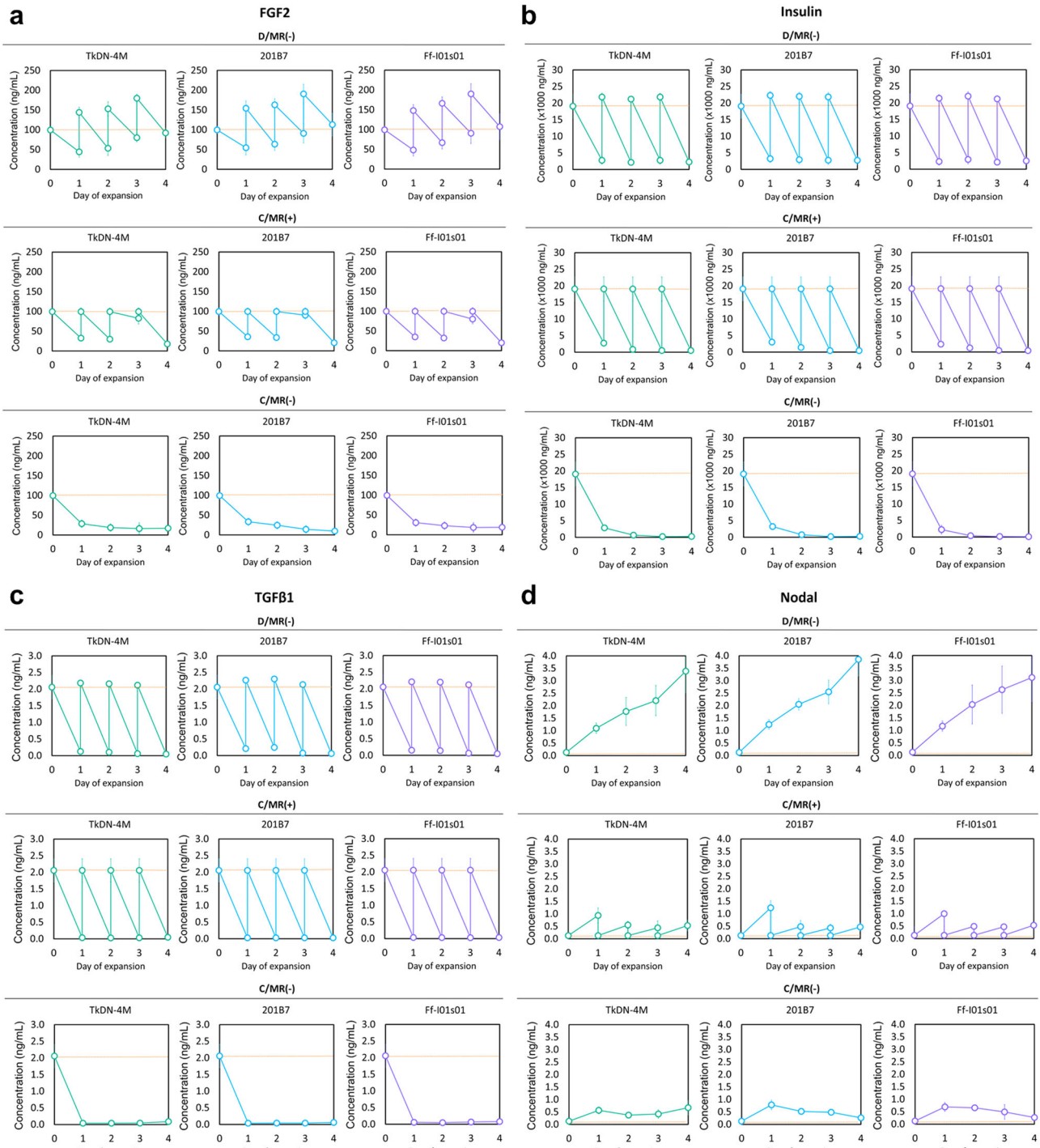

**Fig. 4 The mean concentration of the essential growth factors for maintaining pluripotency in culture compartment during hiPSCs expansion.** The two main growth factors, **a** FGF2 and **b** insulin were well accumulated during the expansion period with D/MR(−) support. **c** Although the TGFβ-1 only accumulated in a low concentration. **d** The endogenous Nodal was able to accumulate and potentially reconstitute the insufficiency of TGFβ-1 in maintaining the pluripotency. Growth-factor concentration in the culture medium is indicated by the red dotted line. (D/MR(–) and C/MR(+): $n = 3$ biologically independent experiments during five passages; C/MR(−): $n = 3$ biologically independent experiments during one passage). Mean ± standard deviation are indicated in each graph.

compensate for the accumulation of toxic metabolic products resulting from cellular metabolism, small molecules, such as lactate, need to be frequently removed from the culture medium when performing high-density culture. Continuous lactate removal using dialysis fed-batch support successfully maintained lactate concentrations in the upper culture compartment below the critical concentration, thereby eliminating the growth-limiting conditions caused by lactate accumulation while supplying glucose. Moreover, we observed an exponential growth curve associated with the maximum cell density and its relationship with low lactate concentration, suggesting that there remains potential for further increases in cell density for this system. Interestingly, a decent amount of glucose remained in conventional suspension culture, even in absence of medium

**Table 1 The medium formulation used in this study.**

| Culture medium component | Molecular weight | Volume or concentration |
|---|---|---|
| 1. DMEM/F12 basal medium (Life Technologies, USA) containing:<br>○ Carbohydrate<br>○ Amino acids<br>○ Inorganic salt<br>○ Vitamins<br>○ Buffer<br>○ Antioxidant<br>○ Other small molecule components | All components <1 kDa | 2 ml (culture compartment); 15 ml (dialysate compartment) |
| 2. Supplemented growth factors (only administrated in culture compartment): | | |
| ○ FGF2 (Life Technologies, USA) | 18 kDa | 100 ng/ml |
| ○ Insulin (Life Technologies, USA) | 5.74 kDa | 19.4 µg/ml |
| ○ Transferrin (Life Technologies, USA) | 80 kDa | 10.7 µg/ml |
| ○ TGFβ-1 (Life Technologies, USA) | 25 kDa | 2 ng/ml |
| 3. FP003 (Nissan chemical, Japan) (only administrated in culture compartment) | 500 kDa | 2% |

replacement during culture (C/MR(−)), although the cell number was extremely depleted relative to the early days of the culture. This suggested that lactate accumulation rather than glucose starvation was the primary limiting factor for cell proliferation in suspension culture. This phenomenon occurs because of hiPSCs exhibit higher anaerobic respiration and metabolism relative to the differentiated cell types. Consequently, lactate secretion was much higher than glucose consumption due to the dependency of PSCs on glycolysis for their energy demands[48]. In addition, this study also revealed a correlation between the increased lactate concentrations, which are likely triggered by medium acidification[49] with the occurrence of DNA damage and genomic alterations in some of the hiPSC lines. Therefore, continuous medium refinement by dialysis is also important to maintain their genomic integrity during the high-density culture.

The inclusion of the FP003 biopolymer was successfully created a low-shear-stress culture environment and prevented hiPSC agglomeration. PSCs exhibit a high tendency to form an aggregate when cultured in suspension[50]; therefore, we utilized dynamic conditions to control the aggregate size and prevent excess agglomeration[2,50–53]. On the other side, the lack of agitation can result in large aggregates with necrotic areas which are often caused by unequal exposure to nutrition and secreted toxic metabolites due to mass-transfer limitations and failure to reach some aggregated cells[54]. However, the excessive mechanical stress resulting from this condition can also potentially induce unwanted spontaneous differentiation and affect cellular viability[52,55]. To prevent the negative effect of this agitation, we added FP003 to the culture medium. Otsuji et al.[19] revealed the potential of FP003 for preventing agglomeration in static hiPSC suspension culture. In the current study, we confirmed that the addition of FP003 did not much affect the mass transfer of macromolecules, such as glucose and lactic acid in our dialysis-culture system. In addition, comparison with rotary suspension culture indicated that FP003 addition was improved cellular viability, possibly due to increased culture medium viscoelasticity via modification of rheological properties. In suspension culture, low-acyl gellan gum in FP003 forms a microfiber-network structure that keeps cells floating in a well-dispersed manner and blocks agglomeration between aggregates (Fig. 1b). As a result, the high-density expansion resulted in a uniform aggregate population with decent growth in size. Moreover, the hiPSCs grew to a tolerable size that allowed the transfer of oxygen, nutrition, and waste products throughout the aggregates, indicated by the absence of necrotic areas inside the aggregates.

The simple dialysis-culture platform shows its potential to save the cost up to 80.31–92.47% from routine normal density suspension culture per unit of produced cells (Supplementary Figure 6). This platform would be useful for evaluating the feasibility of dialysis operations in PSC culture using minimal resources. The complexity of currently available large-scale culture systems makes technical operations during PSC culture challenging to optimize. In addition, medium refinement could overcome the requirement for a high metabolic rate demanded by PSCs in the forms of nutrition transfer, waste-product removal, and growth factors supplementation. This culture system allows various culture conditions simultaneously, enabling optimization of parameters, such as cell metabolism, growth-factor use, device permeation associated with cell growth, pluripotency, and differentiation capacity. Moreover, mathematical simulations describing the metabolic kinetics associated with both exogenous and endogenous growth factors represent useful references when designing larger-scale dialysis-culture systems. Although the multiple units of this simple culture system may be employed in parallel to generate a scalable amount of hiPSCs, a larger culture system with similar principles needs to be developed. However, larger-scale systems, such as slow-turning lateral vessels[9] or stirred-suspension bioreactors with dialysis support[10], might still have several problems. For example, the high cell density expansion required better oxygenation and unwanted side effects of mechanical stress, both of which are broadly design-dependent. Therefore, as the next step, we have been currently trying to scale up the similar rotational culture-based system to 10 mL cell culture compartment supported by a perfusion system and hollow fiber modules. Our current study was limited to the proliferation phase and pluripotency maintenance. Nevertheless, the potential reductions in production costs associated with utilization of both endogenous and exogenous growth factors might contribute to future optimization of larger-scale and cost-effective hiPSC-production methods. Furthermore, such methods could improve both PSC proliferation and differentiation, which require optimal growth-factor and autocrine utilization.

In summary, we described a dialysis-culture system that promotes efficient hiPSC expansion at a high density while maintaining their pluripotency and differentiation capacities by facilitating growth-factor accumulation together with important autocrine factors under a low-hydrodynamic-stress culture environment. This study provides insight into the feasibility of minimum growth-factor usage, which might promote cost reductions in dialysis-based PSC production.

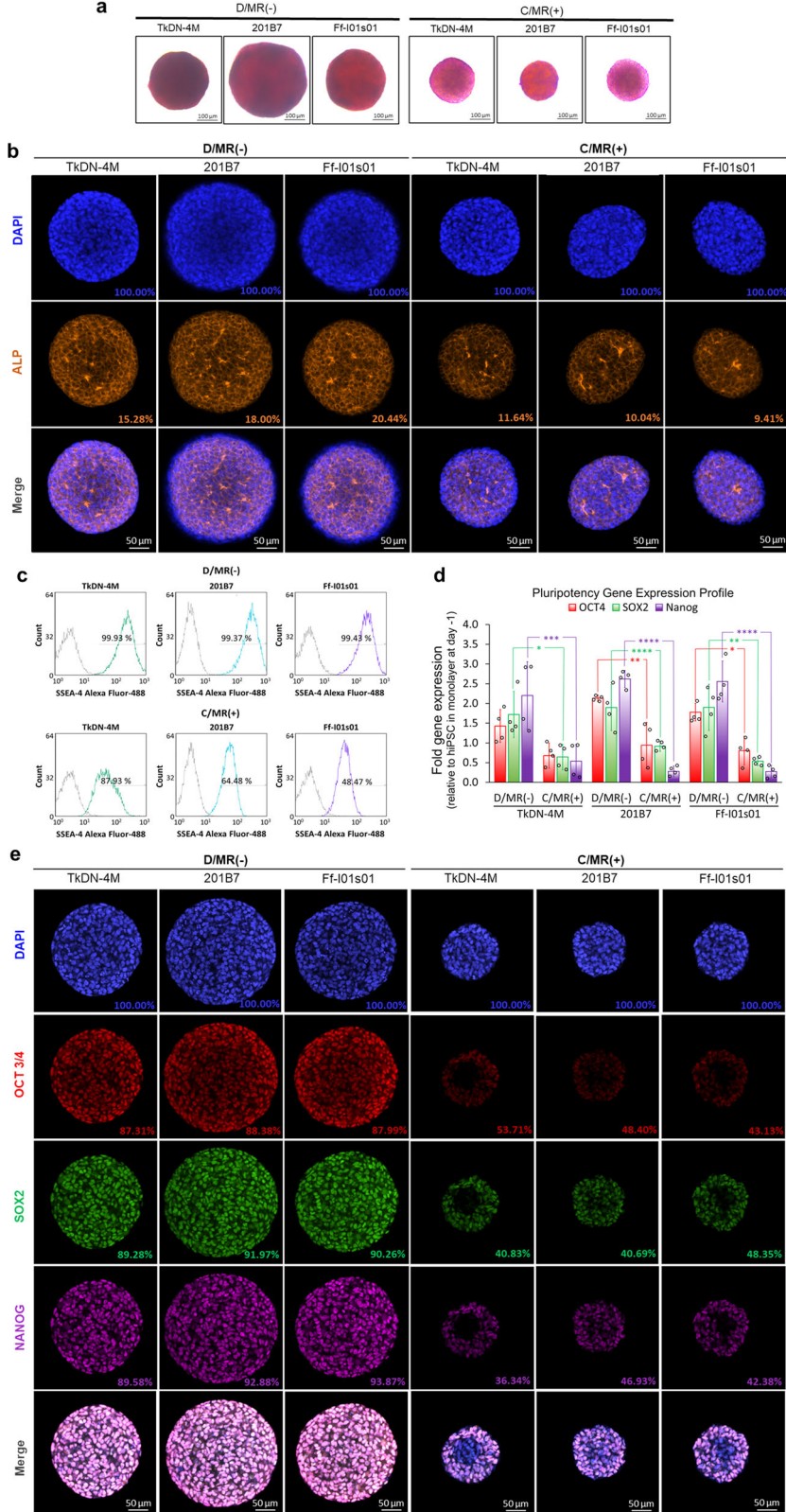

## Methods

**Dialysis device for high-density suspension culture**. The dialysis-culture system included upper and lower dialysate compartments (Fig. 1a). A 40-μm mesh bottom-cell strainer (PluriSelect, Leipzig, Germany) was used as the upper culture compartment, with this insert modified by cutting and removing the bottom mesh layer. To selectively permeate nutrition or waste products, we affixed a 12-kDa MWCO Spectra/Por 4 dialysis membrane (Spectrum Chemical, New Brunswick, NJ, USA) to the bottom side of the strainer using alkyl-α-cyanoacrylate-based surgical-grade tissue adhesive (Aron Alpha A; Daiichi Sankyo, Japan). The upper compartment was then placed in six-deep well plates (Corning, NY, USA) as dialysate-compartment inserts. As a control condition, the cell strainer was directly affixed to the bottom surface of six-well untreated plates (Iwaki, Tokyo, Japan) using Aron Alpha A tissue adhesive (Daiichi Sankyo). All devices were sterilized using an ethylene oxide gas sterilizer before use. The detailed device construction was explained in Supplementary method 1.

**Fig. 5 The characterization of the pluripotent state is evaluated after long-term expansion by different pluripotency markers. a** The hiPSCs expanded in D/MR(−) showing higher alkaline phosphatase (ALP) activity. **b** This activity is also related to higher expression of intracellular ALP in the hiPSCs expanded in D/MR(−), detected by immunocytochemistry. The percentage indicated the protein expression of cytoplasmic *ALP*, relative to nuclear staining (DAPI). **c** The higher expression level of SSEA-4 surface marker and **d** Three main transcription factors regulating pluripotency (*OCT4, SOX2,* and *Nanog*) were detected in hiPSCs expanded in D/MR(−). (*n* = 4 biologically independent experiments). Mean ± standard deviation are indicated in each graph. Statistical significance: ****$p$ < 0.0001; ***$p$ < 0.001; **$p$ < 0.01; *$p$ < 0.05. **e** Their expression in the protein level was confirmed by immunocytochemistry of *OCT3/4, SOX2,* and *NANOG* of the aggregates from the three different hiPSC lines cultured in D/MR(−) and C/MR(+). The percentage indicated the number of positive cells, relative to nuclear staining (DAPI). The C/MR(−) was excluded from the analysis due to the insufficient cell yield.

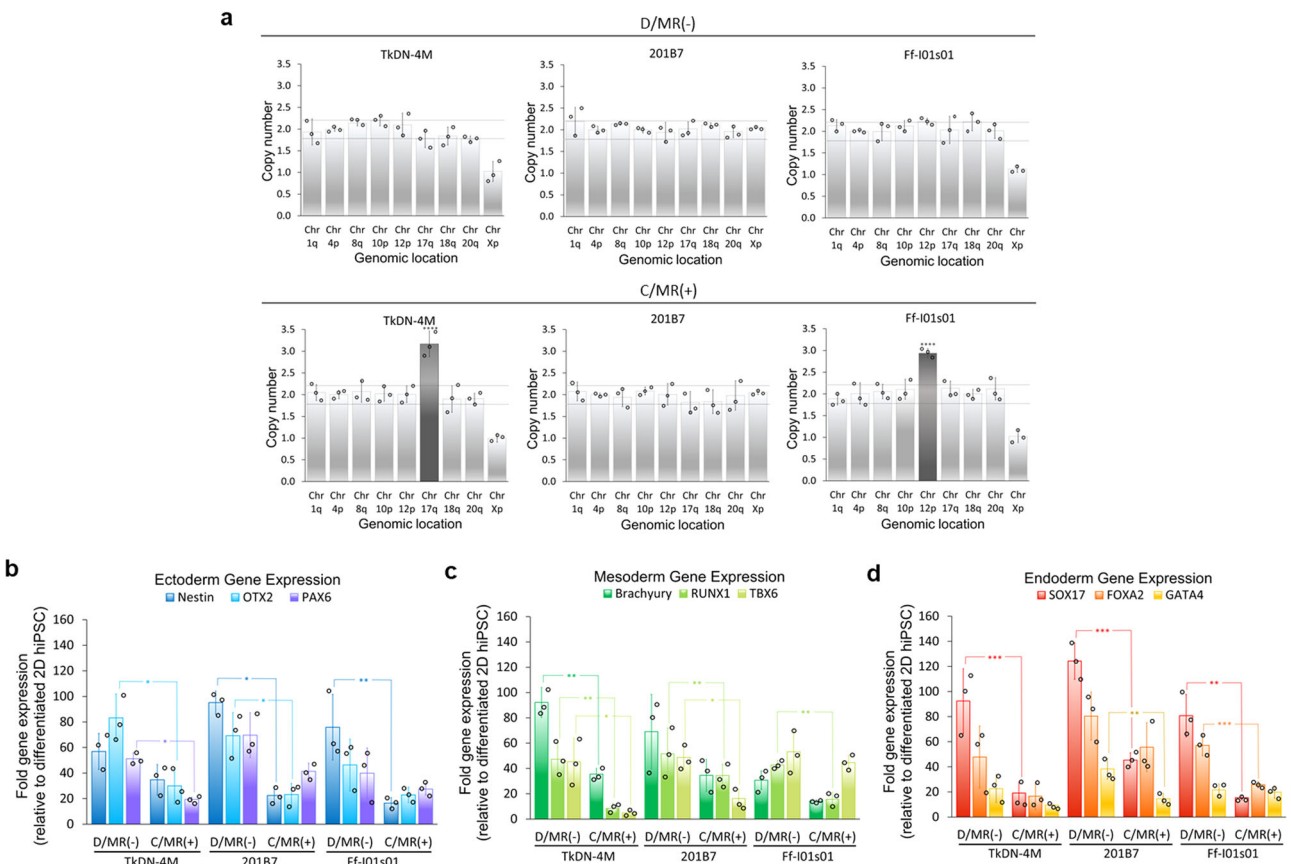

**Fig. 6 Genomic stability and differentiation potential of hiPSCs after long-term culture. a** Chromosomal abnormality was detected in two hiPSCs cultured in C/MR(+) (*n* = 3 biologically independent experiments). A copy number <1.8 or >2.2 (showing by dotted line) with *p* value ≤0.05 in Chr 1q-20q indicates the presence of chromosomal abnormality in the culture (indicated by the black bar). The Chr Xp showed the sex chromosome with normal copy number 2 or 1. The direct differentiation assay was performed to validate the ability to differentiate into three germ lineage. The gene-expression levels of the trilineage marker of the germ layers were evaluated by qPCR: **b** ectoderm (*Nestin, OTX2,* and *PAX6*), **c** mesoderm (*Brachyury, RUNX1,* and *TBX6*), and **d** endoderm (*SOX17, FOXA2,* and *GATA4*). All gene expression was normalized by the differentiated monolayer hiPSCs from day −1. The C/MR(−) culture group without medium replacement was excluded from the analysis due to the insufficient yield after the first passage (*n* = 3 biologically independent experiments). Mean ± standard deviation are indicated in each graph. Statistical significance: ****$p$ < 0.0001; ***$p$ < 0.001; **$p$ < 0.01; *$p$ < 0.05.

**Permeability test in cell-free conditions.** The effect of FP003 on device performance was measured by glucose and lactic acid measurements during a 12 h penetration test under cell-free conditions. Low-glucose DMEM (2 mL; Sigma-Aldrich, St. Louis, MO, USA) containing 0.2% FP003 and 0.8 g/L lactic acids (Sigma-Aldrich) was added to the upper culture compartment, and 15 mL of high-glucose DMEM (Sigma-Aldrich) was placed in the lower dialysate compartment. The dialysis-culture system was then placed in a 120-rpm rotary shaker. We collected 50-μm samples of the medium every 2 h and measured changes in glucose and lactic acid concentration using a YSI 2950 multipurpose bioanalyzer (YSI, Yellow Springs, OH, USA).

To test the ability of the device to accumulate macromolecules in the upper culture compartment and differences in permeation of differently sized molecules, 2 μg/mL FITC (Sigma-Aldrich) at different molecular weights (4, 10, and 20 kDa) was added to the upper culture compartment. This experiment was performed using DMEM basal medium in both compartments, similar to the penetration test for macromolecules. Medium samples were collected after 12 h, and FITC concentration was measured using a Wallac Arvo SX 1420 multilabel counter (PerkinElmer, Waltham, MA, USA).

**Monolayer hiPSC culture.** The TkDN-4M hiPSC line was provided by the Stem Cell Bank, Centre for Stem Cell Biology and Regenerative Medicine, University of Tokyo (Tokyo, Japan)[56]. The 201B7 was provided by the Center of iPS Research and Applications (CiRA), Kyoto University (Kyoto, Japan), and Ff-I01s01 was provided by the Center of iPS Research and Applications Foundation (CiRA Foundation), Kyoto University (Kyoto, Japan). All cells were acclimatized and maintained in vitronectin-coated tissue-culture dishes using a complete supplemented Essential 8 (E8) culture medium (Thermo Fisher Scientific, Waltham, MA, USA) according to manufacturer instructions. The hiPSCs used in this study were cultured in <30 passages.

**LDH release assay.** HiPSC aggregates were formed by inoculation of a $2 \times 10^6$ single-cell suspension/well in six-well plates for 24 h culture in complete medium supplemented with 0.2% free-fatty acid-bovine serum albumin (BSA) in a 90-rpm rotary shaker. The formed aggregate was transferred to dialysis culture system (D/ MR(−)) with or without 0.2% FP003 and cultured with a rotary shaker at 120 rpm. The cells were harvested by sedimentation in 10-mL tubes, and the supernatant was

isolated for LDH analysis using an LDH cytotoxic assay kit (Wako Pure Chemical, Osaka, Japan) according to manufacturer instructions. The culture medium containing released LDH isolated from the single-cell hiPSCs suspension that was intentionally treated with 0.2% Tween 20 was used as a positive control. The luminescence of each sample was measured by Wallac Arvo SX 1420 multilabel counter (PerkinElmer, Massachusetts, USA)

**Caspase 3/7 activity assay**. HiPSC aggregates were formed by inoculating single cells at $2 \times 10^6$/well in six-well plates and culturing in a complete medium supplemented with 0.2% free-fatty acid-BSA on a rotary shaker at 120 rpm for 24 h. The formed aggregates were transferred to the dialysis culture system (D/MR(−)) with or without 0.2% FP003 and cultured on a rotary shaker at 120 rpm for 24 h. Afterward, 1:1 ratio of Caspase-Glo® 3/7 reagent (Promega, Wisconsin, USA) was mixed with the aggregates suspension in a 10 ml conical tube (Corning, USA). The mixture was then moved into six-well plates and rotated at 90 rpm at room temperature for 2 h. The serial number of single-cell hiPSCs suspension that was treated with 200 ng/ml Doxorubicin (Sigma-Aldrich) for 24 h was used as a calibrator. The luminescence of each sample was measured by Wallac Arvo SX 1420 multilabel counter (PerkinElmer, Massachusetts, USA).

**High-density hiPSC suspension culture**. To increase simplicity and reduce costs, the culture medium was divided into two main components. Basal medium included DMEM/F12 (Life Technologies, Carlsbad, CA, USA) supplemented with complex macromolecule nutrition and macromolecule growth factors (Table 1). hiPSC aggregates were formed by inoculation of a $2 \times 10^6$ single-cell suspension/well in six-well plates for 24-h culture in complete medium supplemented with 0.2% free-fatty acid-bovine serum albumin in a 90-rpm rotary shaker. On the day of culture, the dialysis-culture insert was placed in six-deep well plates, and the dialysis membrane was activated by pre-wetting the dialysis layer using 2 mL of sterile $H_2O$ 15 min to 30 min before starting the culture. After removal of the $H_2O$, 15 mL of DMEM/F12 basal medium was added to the lower dialysate compartments. The hiPSC aggregates were harvested and transferred to the upper culture compartment using 2 mL of DMEM/F12 basal medium along with 0.2 % BSA and 0.2% FP003, which were added according to manufacturer instructions to the upper culture compartments on the first day of culture (Fig. 4b). The hiPSCs expansion was performed on a 120-rpm rotary shaker for 4 days with daily replacement of basal medium in the lower dialysate compartment and supplementation with 2% growth factors in groups not undergoing medium replacement in the upper culture compartment. To simplify the operation in D/MR(−), growth-factor supplementation consisting 100 ng/mL FGF2, 19.4 µg/mL Insulin, 10.7 µg/mL Transferrin, and 2 ng/mL TGFβ-1 was added only in the upper culture compartment every day, and the culture medium was not replaced to preserve the growth factors accumulation. (Fig. 1). As a control group, suspension culture was performed using the same volume in a blocked culture insert with (C/MR(+)) or without (C/MR(−)) manual daily complete medium replacement.

**Morphological analysis**. Each of the aggregate groups was moved to 12-well plates, and macroscopic and microscopic images were obtained by light microscope (Olympus, Japan), and aggregate diameter was analyzed using ImageJ 1.53 software (National Institutes of Health, Bethesda, MD, USA).

**Cell counting**. After morphological analysis by light microscopy (Olympus, Tokyo, Japan), the aggregates were collected, centrifuged at 1000 rpm for 3 min, and the supernatant was removed from the tube. To obtain single cells, 1 mL of TrypLE dissociation reagent (Thermo Fisher Scientific) was added to the aggregates and incubated for 10 min to 15 min at 37 °C, followed by homogenization by gentle pipetting. The cells were then diluted 100-fold in phosphate-buffered saline (PBS) and counted using a haemocytometer (Tatai-type; Japan).

**Measurement of glucose and lactate concentrations**. Glucose and lactate concentrations during the 4-day culture in different high-density configurations were measured by collecting 50-µL samples every 24 hand assessment using a YSI 2950 multipurpose bioanalyzer (YSI, Ohio, USA).

**Haematoxylin–eosin staining of cross-sectioned aggregates**. Aggregates were collected and fixed with 4% paraformaldehyde (Wako Pure Chemical) in PBS for 1 h at room temperature, washed in PBS, and cultured in 30% PBS-sucrose solution (Wako Pure Chemical) at 4 °C overnight. The sucrose solution was removed, and the aggregates were placed in a cryomold after embedding with Tissue-Tek OCT compound (Sakura, Alphen aan den Rijn, The Netherlands) at −20 °C until hardened. Thin sections (10 µm) were obtained using a cryostat and mounted onto glass slides for haematoxylin–eosin staining.

**Measurement of growth-factor concentrations**. The FGF2, Insulin, and TGFβ-1 concentrations were measured by Duo Set Elisa Kit (R&D Biosystems, Minessota, USA). A 100-µL solution containing diluted capture antibody was immobilized in each well of 96-well enzyme-linked immunosorbent assay (ELISA) plates and incubated overnight at 37 °C. The plates were then washed with 200 µL of wash

buffer, followed by the addition of 300 µL of blocking buffer (reagent diluent) and incubation for 1 h at 25 °C. 100 µL of each medium sample or standard in reagent diluent was plated in each well and incubated for 2 h at room temperature. The plates were washed with wash buffer, and 100 µL/well of the diluted detection antibody (R&D Biosystems) was added and incubated for 2 h at 25 °C. The plates were then washed with wash buffer, and 100 µL of streptavidin-horseradish peroxidase solution (R&D Biosystems) was added and incubated at room temperature for between 20 min at 25 °C. To obtain a color reaction, the plates were washed with wash buffer, and 100 µL of a substrate solution was added to each well, followed by incubation for 20 min at 25 °C dark places. The color reaction was stopped by adding 50 µL/well stop solution. Fluorescence intensity was measured using a Wallace Arvo SX 1420 multilabel counter (PerkinElmer).

Nodal detection was performed using a human Nodal ELISA kit (LSBio, Seattle, WA, USA) according to manufacturer instructions provided with the kit. Fluorescence intensity was measured using a Wallace Arvo SX 1420 multilabel counter (PerkinElmer, MA, USA). Further information on the ELISA kit used in this analysis is listed in Supplementary Table 1.

**Analysis of the genetic abnormality**. Genomic DNA was isolated using Genomic DNA Purification kit (StemCell Technologies, Vancouver, Canada) followed by qPCR analysis using hPSCs Genetic Analysis kit (StemCell Technologies, Vancouver, Canada) according to manufacturer's instruction. The genomic DNA amplification and analysis were performed by StepOnePlus kit qPCR (Thermo Fisher Scientific, MA, USA). The genetic abnormality was able to be detected when the copy number of Chr 1q-20q is <1.8 or >2.2 with a $p$ value <0.05.

**Directed differentiation assay**. The aggregates were dissociated by 30-min incubation in Accutase solution at 37 °C. The cell suspension was then filtered through a 40-µm cell strainer (Corning) and counted using a hemocytometer (Tatai, Japan). The hiPSCs functional assay was performed by using STEMdiff™ Trilineage Differentiation Kit (StemCell Technologies, Vancouver, Canada) to evaluate their capability to differentiate into three germ layers. For ectoderm induction, $4 \times 10^5$ hiPSCs per well were plated in matrigel coated-24 well plates using STEMdiff™ Trilineage Ectoderm Medium (StemCell Technologies, Vancouver, Canada) supplemented with 10 µM Y-27632. The medium was replaced daily for another 6 days without Y-27632 (Wako, Japan). For mesoderm induction, $1 \times 10^5$ hiPSCs per well were plated in matrigel coated-24 well plates using mTeSR™1 medium (StemCell Technologies, Vancouver, Canada) supplemented with 10 µM Y-27632 (Wako, Japan). The medium was replaced daily by STEMdiff™ Trilineage Mesoderm Medium (StemCell Technologies, Vancouver, Canada) for another 4 days without Y-27632. For endoderm induction, $4 \times 10^5$ hiPSCs per well were plated in matrigel coated-24 well plates using mTeSR™1 medium (StemCell Technologies, Vancouver, Canada) supplemented with 10 µM Y-27632 (Wako, Japan). The medium was replaced daily by STEMdiff™ Trilineage Endoderm Medium (StemCell Technologies, Vancouver, Canada) for another 4 days without Y-27632. The gene-expression levels of the transcription markers of each lineage were analyzed by quantitative real-time PCR.

**Quantitative real-time PCR**. mRNA was isolated using Trizol reagent (Life Technologies) and reverse-transcribed using ReverTra Ace master mix (Toyobo, Osaka, Japan), followed by qPCR analysis using Thunderbird SYBR qPCR mix (Toyobo) according to manufacturer's instructions. Gene amplification was performed using a StepOnePlus kit qPCR (Thermo Fisher Scientific) following the manufacturer's instruction. The primer sequences used in this analysis are listed in Table 2.

**Alkaline phosphatase activity assay**. To detect intracellular alkaline phosphatase, aggregates were harvested and stained with an AP staining kit II (Stemgent, Cambridge, WA, USA), with several modifications for aggregate staining using gentle rotational agitation. After washing with 1× PBST, the aggregates were incubated with 2 mL fixed solution in six-well plates at room temperature for 30 min and then washed with 1× PBST and incubated with 3 mL AP substrate solution in the dark for 90 min at room temperature, followed by washing with 1× PBST. The stained aggregates were observed under a light microscope (Olympus, Japan).

**FACS analysis**. hiPSC aggregates were harvested and dissociated by 30-min incubation in Accutase solution at 37 °C. The cell suspension was then filtered through a 40-µm cell strainer (Corning) and counted using a hemocytometer (Tatai, Japan). In all, $1 \times 10^6$ single cells suspension was permeabilized with 0.5% Triton X for 10 min and fixed with 4% paraformaldehyde for 3 h. Afterwards, the single cells were stained with 1:1000 Alexa Fluor 488 anti-human SSEA-4 Antibody (Biolegend, San Diego, CA, USA). The single cells suspension was washed by 100 rpm centrifugation using PBS. The FACS was performed using Coulter Epics flow cytometer (Beckman Coulter, Brea, CA, USA).

**Immunocytochemistry**. Aggregates were collected and fixed with 4% paraformaldehyde overnight at 4 °C. To preserve the morphology during cross-

**Table 2 The list of primers used in this study.**

| No. | Gene | Forward primer | Reverse primer |
|---|---|---|---|
| 1 | B-Actin | 5′-CCTCATGAAGATCCTCACCGA-3′ | 5′-TTGCCAATGGTGATGACCTGG-3′ |
| 2 | OCT4 | 5′-AGTGGGTGGAGGAAGCTGACAAC-3′ | 5′-TCGGTTGTGCATAGTCGCTGCTTGA-3′ |
| 3 | SOX2 | 5′-GGCAGCTACAGCATGATGCAGGAGC-3′ | 5′-CTGGTCATGGAGTTGTACGCAGG-3′ |
| 4 | Nanog | 5′-AGGACAGGTTTCAGAAGCAGAAGT-3′ | 5′-TCAGACCATTGCTAGTCTTCAACC-3′ |
| 5 | OTX2 | 5′-GGAGAGGACGACATTTACTAGG-3′ | 5′-TTCTGACCTCCATTCTGCTG-3′ |
| 6 | Nestin | 5′-GCGTTGGAACAGAGGTTGGA-3′ | 5′-TGGGAGCAAAGATCCAAGAC-3′ |
| 7 | PAX6 | 5′-GAGTGCCCGTCCATCTTTG-3′ | 5′-GTCTGCGCCCATCTGTTGCTTTTC-3′ |
| 8 | Brachyury | 5′-ATCGTGGACAGCCAGTACGA-3′ | 5′-GCCAACTGCATCATCTCCAC-3′ |
| 9 | RUNX1 | 5′-CCCTAGGGGATGTTCCAGAT-3′ | 5′-TGAAGCTTTTCCCTCTTCCA-3′ |
| 10 | TBX6 | 5′-AAGTACCAACCCCGCATACA-3′ | 5′-TAGGCTGTCACGGAGATGAA-3′ |
| 11 | SOX17 | 5′-AGGAAATCCTCAGACTCCTGGGTT-3′ | 5′-CCCAAACTGTTCAAGTGGCAGACA-3′ |
| 12 | FOXA2 | 5′-GCATTCCCAATCTTGACACGGTGA-3′ | 5′-GCCCTTGCAGCCAGAATACACATT-3′ |
| 13 | GATA4 | 5′-AGCACACTGCATCTCTCCTGTG-3′ | 5′-CTCCGCTTGTTCTCAGATCCTC-3′ |

sectioning, the aggregates were incubated with an 8% sucrose solution overnight following sample fixation in a cryomold using Tissue-Tek OCT compound (Sakura) and freezing for at least 6 h at −80 °C. A 20-μm thin cross-section was obtained using a Leica CM1850 cryostat (Leica Biosystems, Wetzlar, Germany) and fixed on poly-L-lysine-coated slides (Sigma-Aldrich, Missouri, USA). The sections were then stained using 1:10 dilution of Human Pluripotent Stem Cell 3-Color Immunocytochemistry Kit (R&D Biosystems, USA) in blocking buffer overnight in 4°C and counterstained with 1:1000 nuclear staining solution DAPI (Dojindo, Kumamoto, Japan) at room temperature for 10 min. The section washed with PBS and fluorescence microscopy image was performed by FV3000 Confocal Laser Scanning Microscope (Olympus, Tokyo, Japan).

To identify the cytoplasmic alkaline phosphatase, the sections were incubated with 10 μg/mL Mouse anti-Human Alkaline Phosphatase/ALPL Antibody (R&D Biosystems) overnight at 4°C. After washing with PBS, the sections were then incubated with Goat Anti-Mouse IgG Alexa Fluor 488 secondary antibody at room temperature for 3 h and counterstained with 1:1000 nuclear staining solution DAPI (Dojindo, Kumamoto, Japan) for 10 min. The section washed with PBS and fluorescence microscopy image was performed by FV3000 Confocal Laser Scanning Microscope (Olympus, Tokyo, Japan). The detailed information about the antibody used in this study is listed in Supplementary Table 1.

**Statistics and reproducibility**. The data were obtained from at least three independent experiments using three different hiPSCs lines. Statistical analysis was performed using GraphPad Prism software (v.8.3.0; GraphPad Software, San Diego, CA, USA). Statistical significance was determined by one-way analysis of variance with Tukey's multiple comparison test. A $p < 0.05$ was considered significant.

**Reporting summary**. Further information on research design is available in the Nature Research Reporting Summary linked to this article.

## Data availability

The source data for graphs and charts in this study are available as Supplementary data 1. Further remaining information can be obtained from the corresponding author upon reasonable request.

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

## Acknowledgements

This study was a part of a research project for the development of next-generation islet transplantation using iPS cells (leading by Professor Atsushi Miyajima at The University of Tokyo, Japan), supported by the Research Center Network for Realization of Regenerative Medicine, Japan Agency for Medical Research and Development (AMED).

## Author contributions

Y.S., I.H. and F.G.T. designed the experimental project. F.G.T. and Q.Y.L. performed experiments and analyzed the data. M.H. and M.M. provided the FP003. T.M. provide the culture conditioning support for hiPSC line. F.G.T., I.H., Y.S., M.I. and Y.K. designed the dialysis device. F.G.T., Q.Y.L., I.H., M.N., Y.S. wrote the manuscript.

## Competing interests

The authors declare no competing interests.
