## [Transparent Peer Review File · Communications Biology]

Reviewers' comments:

Reviewer #1 (Remarks to the Author):

I have read this manuscript 'High-density hiPSCs expansion supported by growth factors accumulation in a simple dialysis-culture platform' by Fuad Gandhi Torizal and colleagues with interest. I found the introduction very well written, I would only suggest that reference to work from Nicola Elvassore group work would be important as the interplay between secreted and added factors is well present in their studies.

The manuscript appears solid to me and with no major conceptual flaws and with original conclusions to my understanding. Results would be of relevance for readers and the authors bridge well biological and engineering concepts reflecting on the value of this method for cell production.

Outstanding features that in my view would improve this manuscript are below. Hope these comments become helpful in improving the authors' manuscript. My major problem is the order of figures in text is bumpy and there is a lack of consistency in how the conditions are referred to. Importantly, I could not find Figure 1H in the results text (!). In Figure 1H, it is important to reword and clarify what 'injured cells' mean clarifying what markers are used for what biological phenomenon (apoptosis?).

Fig 2A the different rotation 90 rpm and the 6 wells, the authors should motivate why these parameters are not controlled separately if not possible to provide data that control for this separately – Fig 4D the finding of Nodal concentration increasing in the system is interesting and one of the main point of the paper. For clarity, it would help if there is consistence in the graphs (although I understand Nodal has not been replenished) and more clarity in the acronyms describing the different conditions (with colour consistency for example).

Just before figure 5A in the text 'better pluripotency' requires rewording. The sizes of the spheroid in Fig 5A are comparable but in Fig 5D, the one without dialysis is smaller it would be important to check pluripotency markers on comparable sizes if feasible. The staining seems really good but it would be good to see a higher resolution figure. It is important to understand the extent to which this approach could be applied to self-renewing condition expansion of human pluripotent stem cells in bioreactors or to formation of embryoid bodies at scale.

Reviewer #2 (Remarks to the Author):

Although human pluripotent stem cell culture is routine, daily media changes, high cost of growth factors required in pluripotent cell media, and reproducibility of differentiation are an obstacle for the mass production of hPSCs. In this manuscript, the authors describe a cost-effective dialysis-culture system for human pluripotent stem cells (hPSCs) at a high density. It is of particular interest for the application of hiPSCs in for example pharmacogenomics which requires the culture of large quantities of hiPSCs.

Major issues

- Daily media changes are an issue for the culture of large quantities of hiPSCs. It is not clear from the text how frequently authors change media in the described dialysis-culture system. Authors should indicate this.

- Along the same line, authors have to show for how many passages they can propagate hiPSCs at the high density in the dialysis-culture system. Besides, authors need to validate the quality of iPSC, at the early and late passage (expression of pluripotency markers and genome stability).
- Authors should have more than one hiPSC line to demonstrate the usability of their cell culture system.
- The authors claim they have developed cost-effective cell culture method. However, they did not present the cost estimate. They need to calculate savings in comparison to the conventional cell culture method.
- Reproducibility of differentiation is one of the main issues with hiPSC culture system. Authors have used spontaneous differentiation, embryoid body assay (EB) to address this issue. However, EB differentiation is highly variable and is not suitable for this purpose. The authors should assess the pluripotency by direct differentiation of hiPSC lines into all three germ layers (there are commercially available trilineage differentiation kits). The result should be compared to conventionally cultured isogenic hiPSC lines, preferably in more than one hiPSC line.

Minor comments

- Line 89-90. Authors claim that FP003 improves pluripotency, which is incorrect. They based this assumption on the reduced lactate dehydrogenase (LDH) leakage assay and Oct4 expression (supplementary). Authors should instead demonstrate that FP003 improves cell survival. LDH leakage assay is useful to indicate necrosis. Authors should perform an additional assay to distinguish whether any detected LDH leakage is due to secondary necrosis from apoptotic cells, such as caspase activity or DNA strand breaks assays.
- Check the figure referencing in the text. Figure 1B lack the reference in the manuscript. Line 89 Figure 1F should be Figure 1H.
- Figure 1H Authors should open-up all abbreviations used in the figures. For example, abbreviation 2D, is not explained.
- Figure 4D. Abbreviation E8 needs to be open-up. Besides D/MR(-), C/MR(+), C/MR(-) needs to be open in the main text and figures.
- Authors should check and correct references, there seem to be some issues with the program they have used. For example, in line 120 there is reference number 1919,20.

Reviewer #3 (Remarks to the Author):

Review of Torizal et al. „High-density hiPSCs expansion supported by growth factors accumulation in a simple dialysis-culture platform “

The manuscript by Torizal et al. demonstrates a novel dialysis-based culture system for human pluripotent stem cells. They provide data which suggests that the use of this cultivation approach reduces cytokine costs and leads to significantly higher cell yields compared to conventional cultures. Moreover, the authors speculate that this system leads to an improved self-conditioning of the medium which improves the culture quality with respect to proliferation, pluripotency marker expression and differentiation potential.

Overall, the article appears technically sound. However, there are some important points that need to

be improved before the manuscript can be considered for publication.

Major points:

- In order to confirm that this protocol is widely applicable, it is mandatory to reproduce key findings (proliferation, pluripotency markers, differentiation potential) with two additional hPSC lines.
- I strongly suggest to provide some long-term cultivation data covering at least 5 passages to demonstrate the robustness of the system
- The authors need also to provide evidence whether the karyotype of the hiPSCs remains stable after long-term cultivation in their system.
- Figure 1: beyond the graphical overview of the dialysis-based culture system I am missing additional information on how the setup looks like and is assembled in real life. To address this issue, some supplementary photos or short video sequences of the setup and how to assemble it are needed.
- Figure 5: The data on pluripotency markers shown in Fig. 5A and D need to be substantiated by quantifications derived from ICC or, alternatively, WB analyses
- The authors make a great deal about saving costs for growth factors. What is the actual savings potential (in %) per unit of produced cells when using E8 medium? This information should be included in the discussion.
- The discussion should also include a paragraph on the scalability of the system.

Minor points:

- Please show data on mass-transfer capability between the two compartments in the presence of FP003 as supplementary data (so far it is "data now show").
- Please explain how the LDH cytotoxic assay kit can determine the "number of injured cells" as stated in the diagram of Figure 1H (the text wrongly refers to Figure 1F by the way!).
- Line 89, "This showed that FP003-mediated decreases in mechanical stress improved pluripotency": This statement is not correct as it refers to the LDH assay which tests for viability.
- Suppl. Figure 1: N=?, is this difference statistically significant? Moreover, a single marker on RNA level not sufficient to claim improved pluripotency.
- Fig. 1: Please provide a simple legend stating the color code of the graphs.
- Fig. 2D: What culture configuration(s) is/are shown here?
- Line 106: I did not understand why proliferation rate was given as 5×10^6 cells/mL, please clarify.
- Figure 3: What cell numbers were used for the glucose/lactate assays in this Fig.? Please state in Figure legend.
- Figure 4: What does the dashed blue line mean? Please state in Figure legend.
- Line 167, "This population exhibited better gene-expression ...": I suggest to change to "This population exhibited higher gene-expression ..." to phrase it in nonjudgmental way.
- Line 171: These aggregates are not "organoid-like" but rather simply embryoid bodies.
- Figure 6A: What is the point of this figure? Please explain. Moreover, it is hard to tell a difference and no quantitative data is provided.
- Figure 6B: Diagram shows OTX1 but legend refers to OTX2. Please clarify.
- Figure 7: My expectation would be that a final figure tries to summarize the key findings of a study. Since this is barely the case here, thus please consider to move it to the Supplement. Please check also the grammar of the legend ("... of The Nanog regulation ...").

Answer to the Reviewer # 1

I have read this manuscript 'High-density hiPSCs expansion supported by growth factors accumulation in a simple dialysis-culture platform' by Fuad Gandhi Torizal and colleagues with interest. I found the introduction very well written, The manuscript appears solid to me with no major conceptual flaws and with original conclusions to my understanding. Results would be of relevance for readers and the authors bridge well biological and engineering concepts reflecting on the value of this method for cell production. Outstanding features that in my view would improve this manuscript are below. Hope these comments become helpful in improving the authors' manuscript.

Answer : Thank you very much for your careful reviewing and giving the important suggestion. We believe that your comment and suggestions can improve our manuscript. We revised our manuscript and indicated the correction **in red**, and we also answered your comments below. To simplify the results and avoid confusion when using three hiPSC lines with too much experimental group, we choose the best inoculation density based on our previous experimental data in TkDN-4M hiPSC line. This supportive data was moved to supplementary material as evidence of improvement of cell quality in higher density. We sincerely hope that the revised article and answers make you satisfied.

Reviewers` comments and suggestions :

1. *I would only suggest that reference to work from Nicola Elvassore group work would be important as the interplay between secreted and added factors is well present in their studies.*

Answer: Thank you very much for suggesting additional references. This reference was provided additional information in our revised manuscript. We also included the reference on the third paragraph of discussion section our revised manuscript.

2. *My major problem is the order of figures in text is bumpy and there is a lack of consistency in how the conditions are referred. In Figure 1H, it is important to reword and clarify what 'injured cells' mean clarifying what markers are used for what biological phenomenon (apoptosis?). To. Importantly, I could not find Figure 1H in the results text (!).*

Answer: We are sorry for the figures' order and lack of consistency. In this revised manuscript, we rearrange the figure set and clarify the LDH leakage as secondary apoptosis or necrosis markers instead of “injured cells” in Figure 1H. In addition, we also adding the Caspase 3/7 assay to determine the occurrence of primary apoptosis in Figure 1I. The result showed that most of the non viable cells was tend to experience necrotic rather than apoptosis in mechanical stress resulted from the high density dynamic suspension culture. However, the impact can be significantly reduced by FP003 inclusion in the culture medium. All of the results were calibrated to their value in intentionally necrotic- or apoptotic-induced hPSCs.

3. *Fig 2A the different rotation 90 rpm and the 6 wells, the authors should motivate why these parameters are not controlled separately if not possible to provide data that control for this separately*

Answer: Thank you for your suggestion. The aggregates forming was performed in the 6 well plates at 90 rpm was essential to produce a homogenous aggregates, because the excess agglomeration was often occurred when we directly perform the aggregates forming in the dialysis insert. Afterward, the aggregates

moved into smaller dialysis culture insert with higher rotational speed. The 120 rpm rotational speed are necessary to maintain a proper centrifugal force for aggregates dispersion which affected by lower diameter of the culture insert. We include this explanation in the first paragraph of “improved cell proliferation and aggregate morphology” sub section of Results section.

4. *Fig 4D the finding of Nodal concentration increasing in the system is interesting and one. For clarity, it would help if there is consistence in the graphs (although I understand Nodal has not been replenished) and more clarity in the acronyms describing the different conditions (with color consistency for example).*

Answer: Thank you for the suggestion. In the revised manuscript, we presented the new graph where the daily concentration of Nodal was included in a consistent with the other growth factor measurements in Figure 4D. The result showing that the Nodal was significantly accumulated in simple dialysis culture compartment, when its accumulation was limited by conventional complete medium replacement.

5. *Just before figure 5A in the text 'better pluripotency' requires rewording.*

Answer: We agree with the reviewer. In the revised manuscript, we replace the improper wording “better pluripotency” with “higher level of various pluripotency marker” in the first paragraph of “High-density culture improves pluripotency and differentiation potential” sub section of result section.

6. *The sizes of the spheroid in Fig 5A are comparable but in Fig 5D, the one without dialysis is smaller. It would be important to check pluripotency markers on comparable sizes if feasible. The staining seems really good but it would be good to see a higher resolution figure. It is important to understand the extent to which this approach could be applied to self-renewing condition expansion of human pluripotent stem cells in bioreactors or to formation of embryoid bodies at scale.*

Answer: We are sorry for the confusion caused by the inconsistent number of sample analyzed in previous figures. In previous figure 5A, the size between the aggregates propagate in lower initial cell density are comparable. However, in figure 5D we only provide the spheroid cultured with highest inoculum density (4×10^6 cells/ml) which is consistent with previous figure 5A. In the revised manuscript, we consistently using this density in all of the experimental group. Unfortunately, we found that the aggregates in D/MR(-) population was showed uniformly larger aggregates compared to C/MR(+) (Figure 5A and 5E). We also enlarge and maximize the staining figures size based on space availability in Figure 5E.

Answers to the Reviewer # 2

Although human pluripotent stem cell culture is routine, daily media changes, high cost of growth factors required in pluripotent cell media, and reproducibility of differentiation are an obstacle for the mass production of hPSCs. In this manuscript, the authors describe a cost-effective dialysis culture system for human pluripotent stem cells (hPSCs) at a high density. It is of particular interest for the application of hiPSCs in for example pharmacogenomics which requires the culture of large quantities of hiPSCs.

Answer: Thank you for your kind review and comments, which significantly improve our manuscript. According to your comments, we improve the manuscript and indicated the correction in red. To simplify the results and avoid confusion when using three hiPSC lines with too much experimental group, we choose the best inoculation density based on our previous experimental data in TkDN-4M hiPSC line. We also answered your comment below. We sincerely hope that the revised manuscript makes you satisfied.

Major comments and suggestions

1. *Daily media changes are an issue for it is not clear from the text how frequently authors change media in the described dialysis-culture system. Authors should indicate this.*

Answer: Thank you for the suggestion. Besides in the methods section, we indicate the method of daily basal medium replacement and growth factors supplementation in Figure 2A and include the explanation in the figure legend.

2. *Along the same line, authors have to show for how many passages they can propagate hiPSCs at the high density in the dialysis-culture system. Besides, authors need to validate the quality of iPSC, at the early and late passage (expression of pluripotency) markers and genome stability.*

Answer: We thank the reviewer for suggesting a very important point to improve the manuscript. In this revised manuscript, we successfully extended the propagation up to 5 passages. The quality of expanded hiPSCs were validated by their expression levels of pluripotency markers, compared with those before the expansion (Figure 5D). The hiPSCs expanded in dialysis device was also be able to maintain their genome stability during 5 passages in high-density culture (Figure 6A)

3. *Authors should have more than one hiPSC line to demonstrate the usability of their cell culture system.*

Answer: The reviewers address an important point. In view of actual applications, our system need to demonstrate usability, regardless of lines of hiPSCs. In our revised manuscript, we added two additional hiPSC lines, 201B7 (the common hiPSCs cell line that used in Japan) and Ff-I01s1 (the clinical hiPSC line established by CiRA) to demonstrate the usability in multiple hiPSC lines. We embedded the data for this hiPSC line in Figure 1-5 in the revised manuscript.

4. *The authors claim they have developed cost-effective cell culture method. However, they did not present the cost estimate. They need to calculate savings in comparison to the conventional cell culture method.*

Answer: Thank you for the suggestion. We also think the cost estimation are very important to show the realistic advantage of this culture system for actual utilization. We provide the actual cost saving of our high density culture (D/MR(+)) with both high-density culture in conventional daily medium replacement

(C/MR(-)) and the routine conventional suspension culture with normal inoculation density (ND) in Supplementary figure 6 and described in the sixth paragraph of discussion section.

5. *Reproducibility of differentiation is one of the main issues with hiPSC culture system. Authors have used spontaneous differentiation, embryoid body assay (EB) to address this issue. However, EB differentiation is highly variable and is not suitable for this purpose. The authors should assess the pluripotency by direct differentiation of hiPSC lines into all three germ layers (there are commercially available trilineage differentiation kits). The result should be compared to differentiated cultured isogenic hiPSC lines, preferably in more than one hiPSC line.*

Answer: We agree with the reviewers` suggestion. In this revised manuscript, we assess the pluripotency by performed direct trilineage differentiation assays of the hiPSC cell after 5 time passages using the kit recommended by reviewer. We evaluated the gene expression level of each lineage markers, normalized to its expression on each differentiated hiPSC line from day -1 (before expansion). The results was presented in Figure 6B, 6C, and 6D and described in the third paragraph of “High-density culture improves pluripotency and differentiation potential” sub section in the results section.

Minor comments and suggestions

1. *Line 89-90. Authors claim that FP003 improves pluripotency, which is incorrect. They based this assumption on the reduced lactate dehydrogenase (LDH) leakage assay and Oct4 expression (supplementary). Authors should instead demonstrate that FP003 improves cell survival. LDH leakage. Assay is useful to indicate necrosis. Authors should perform an additional assay to distinguish whether any detected LDH leakage is due to secondary necrosis from apoptotic cells, such as caspase activity or DNA strand breaks assays.*

Answer: We thank the reviewer for the suggestion. As suggested, in this part of the revised manuscript, we are focusing on cell survival improvement by the inclusion of FP003. We removed the previous OCT4 expression and more focused on cell viability assays. We repeated the LDH assay to determine necrotic/secondary apoptosis (1H) and caspase 3/7 activity to determine the primary apoptosis (Figure 1I). Wealso confirmed that the mechanical stress tend to induced cellular necrosis rather than apoptosis (Figure 1H and 1I).

2. *Check the figure referencing in the text. Figure 1B lack the reference in the manuscript. Line 89 Figure 1F should be Figure 1H.*

Answer: We apologize for our referencing error. We fixed the referencing errors in the current manuscript, including Figure 1H.

3. *Figure 1H Authors should open-up all abbreviations used in the figures. For example, abbreviation 2D, is not explained.*

Answer: Thank you for the suggestions. We open up all abbreviations in the manuscript. To make it clearer, we also replaced the term “2D” with “monolayer”

4. *Figure 4D. Abbreviation E8 needs to be open-up. Besides D / MR (-), C / MR (+), C / MR (-) needs to be open in the main text and figures.*

Answer: Thank you for the suggestions. We open up all abbreviations of E8, D / MR (-), C / MR (+), C / MR (-) in the figure description and the main text of the revised manuscript.

5. *Authors should check and correct references, there seem to be some issues with the program they have used. For example, in line 120 there is reference number 1919,20.*

Answer: We apologize for this citation error. In our revised manuscript, we fixed the citation errors.

Answers to the Reviewer # 3

The manuscript by Torizal et al. Demonstrate a novel dialysis-based culture system for human pluripotent stem cells. They provide data which suggests that the use of this cultivation approach reduces cytokine costs and leads to significantly higher cell yields compared to conventional cultures. Moreover, The authors speculate that this system leads to an improved self-conditioning of the medium which improves the culture quality with respect to proliferation, pluripotency marker expression and differentiation potential. Overall, the article appears technically sound. However, there are some important points that need to be improved before the manuscript can be considered for publication.

Answer: Thank you very much for your careful reviewing and giving the important suggestion. We believe that your comment and suggestions can improve our manuscript. We revised our manuscript and indicated the correction **in red**, and we also answered your comments below. To simplify the results and avoid confusion when using three hiPSC lines with too much experimental group, we choose the best inoculation density based on our previous experimental data in TkDN-4M hiPSC line. We sincerely hope that the revised article and answers make you satisfied.

Major comments and suggestions

1. *In order to confirm that this protocol is widely applicable, it is mandatory to reproduce key findings (proliferation, pluripotency markers, differentiation potential) with two additional hPSC lines.*

Answer: The reviewer addresses an important point. The usability of this system can be better demonstrated in multiple hiPSC lines. In the current manuscript, we adding experimental results from two additional hiPSC cell lines, 201B7 and Ff-I01s01. We presented the proliferation profile in the Figure 2, Pluripotency marker in Figure 5, and differentiation marker in Figure 6. We also describe these findings in the result and discussion section.

2. *I strongly suggest to provide some long-term cultivation data covering at least 5 passages to demonstrate the robustness of the system.*

Answer: Thank you for the suggestion, in this revised manuscript we provide and integrate all analysis of 5 cultivation passages data from three cell lines (TkDN-4M, 201B7, and Ff-I01s01). During 5 passages, all of the hiPSC line can be efficiently proliferated in a good culture environment in terms of nutrition availability and low toxic metabolic product with utilization of accumulated growth factors (Figure 2-4). As a results, the hiPSC line expanded in simple dialysis system showing higher pluripotency marker and differentiation capability without any genomic abnormality (Figure 5-6).

3. *The authors need also to provide evidence whether the karyotype of the hiPSCs remains stable after long-term cultivation in their system.*

Answer: We thank the reviewer for the suggestion. We added the genetic analysis to check the genomic stability during the expansion period in Figure 6A. The results showing that the genomic stability can be preserved in the high density culture of hiPSCs in simple dialysis culture system. In contrary, we found that

the occurrence of genetic abnormality was detected in two hiPSC line (TkDN-4M and Ff-I01s01) cultured in conventional medium replacement, which may caused by high lactate concentration.

4. *Figure 1: beyond the graphical overview of the dialysis-based culture system I am missing additional information on how the setup looks like and is assembled in real life. To address this issue, some supplementary photos or short video sequences of the setup and how to assemble it are needed.*

Answer: Thank you for the suggestion. We added the actual figures which represented the assembling methods of the dialysis culture system and the device for control group in Supplementary method 1 and 2.

5. *Figure 5: The data on pluripotency markers shown in Fig. 5A and D need to be substantiated by quantifications derived from ICC or, optionally, WB analyses.*

Answer: In the revised manuscript, we quantified the protein level of alkaline phosphatase (ALPL) by FACS (Figure 5B). We also quantified the percentages of cells stained positive with pluripotency markers based on the ICC image analysis (Figure 5E). The result showing that based on the quantification, the percentage of positive cells for pluripotency marker was found higher in the hiPSC expanded in simple dialysis culture system.

6. *The authors make a great deal about saving costs for growth factors. What is the actual savings. potential (in%) per unit of produced cells when using E8 medium? This information should be included in the discussion.*

Answer: Thank you for the suggestion. We added the % potential cost reduction in supplementary figure 5 and include the information in the discussion section. The result showing that based on actual cost saving calculated by the total growth factors usage and the volume of medium, the high density culture in simple dialysis system showing a very high cost production efficiency, approximately 80.31-92.42 % (varies among cell line) compared to the routine normal density culture.

7. *The discussion should also include a paragraph on the scalability of the system.*

Answer: Thank you for the suggestion. We mentioned the potential scalability of our culture system in 6th paragraph of discussion section. Although the multiple unit of our simple dialysis device may be employed to generate a scalable amount of hiPSCs, a larger culture system with similar principles need to be designed. Furthermore, several problems such as oxygenation, mechanical stress, and complexity in operations need to be addressed. To improved the scalability, we currently designed a larger system with hollow fiber and perfusion support with up to 10 ml of culture compartment volumes.

Minor comments and suggestions:

1. *Please show data on mass-transfer. capability between the two compartments in the presence of FP003 as supplementary data (so far it is “data now show”).*

Answer: Thank you for the suggestion. We provide the additional glucose-lactate mass transfer between the inclusion and exclusion of FP003 that previously excluded in Figure 1B and 1C in our revised manuscript.

2. *Please explain how the LDH cytotoxic assay kit can determine the “number of injured cells” as stated in the diagram of Figure 1H (the text wrongly refers to Figure 1F by the way!).*

Answer: We are sorry for the wrong figure reference. In our revised manuscript, we corrected the reference and clarify the LDH leakage as secondary apoptosis or necrosis markers. All of the results was normalized

to their value of intentionally necrosis-induced hPSCs in 24 hours. We include this information in the description of figure 1H.

3. *Line 89, “This showed that FP003- mediated decreases in mechanical stress improved pluripotency ”: This statement is not correct as it refers to the LDH assay which tests for viability.*

Answer: Thank you for the important correction. We removed the statement in the latest manuscript version. Instead, we perform LDH assay and caspase 3/7 assay to evaluate the effectiveness of FP003 in reducing the cell death and also determine the prominent mechanisms (apoptosis or necrosis) that involved in the mechanical stress (Figure 1H and 1I). We also removed OCT4 expression in previous Supplementary figure 1 to focused more on the viability assay.

4. *Fig. 1: Please provide a simple legend stating the color code of the graphs.*

Answer: Thank you for the suggestion. We added a simple indicator of each color in the latest graph in Figure 1.

5. *Fig. 2D: What culture configuration (s) is / are shown here?*

Answer: We are sorry for the missing information due to software error. In our revised manuscript, we added an indicator for each culture configuration in Figure 2D.

6. *Line 106: I did not understand why proliferation rate was given as 5×10^6 cells / mL, please clarify.*

Answer: Thank you for the comment. The daily medium replacement in C/MR(+) was not enough to further support proliferation due to lactate production level in 24 hours. As the consequence, the cells were constantly remain at 5×10^6 cells/ml density until the end of the expansion period. To clarify we added the description of the correlation between lactate accumulation in the third paragraph of “*Continuous nutrient supply and toxic metabolite removal*” sub section.

7. *Figure 3: What cell numbers were used for the glucose / lactate assays in this Fig.? Please state in Figure legend.*

Answer : We are sorry for the missing information. We added the information of the cell number in the figure (currently, this density dependent-expansion figure was moved into supplementary figure 2).

8. *Figure 4: What does the dashed blue line mean? Please state in Figure legend.*

Answer: we are sorry for missing the information in the figure legend. In the latest manuscript, we added the information in the legend section, in both new figure 4 and supplementary figure 3.

9. *Line 167, “This population exhibited better gene-expression... ”: I suggest to change to “ This population exhibited higher gene-expression... ” to phrase it in nonjudgmental way.*

Answer : We thank the reviewer for the suggestion. We rephrased the sentence based on this suggestion in our revised manuscript into “exhibited higher level of various pluripotency marker” in the first paragraph of “High-density culture improves pluripotency and differentiation potential” sub section

10. *Line 171: These aggregates are not “ organoid-like ” but rather simply embryoid bodies.*

Answer: We agree with the reviewer. Because the differentiation of embryoid bodies differentiation often produces high variability, we removed the figure and performed the direct differentiation assay which can evaluate the differentiation potential more accurately. The results were presented in Figure 6B,6C, and 6D with the description in the third paragraph of “High-density culture improves pluripotency and differentiation potential” sub section.

11. *Figure 6A: What is the point of this figure? Please explain. Moreover, it is hard to tell a difference and no quantitative data is provided.*

Answer: We agree with the reviewer. Because the assay using spontaneous differentiation of embryoid bodies is highly variable, we replaced the assay with the directed differentiation into all three germ layers using the trilineage differentiation kit to assess the differentiation capability by analysing each expression patterns of gene marker by qPCR (Figure 6A, 6B, and 6C) in the revised manuscript.

12. *Figure 6B: Diagram shows OTX1 but legend refers to OTX2. Please clarify.*

Answer: We apologize for the typing error. We corrected the typing error in the diagram of Figure 6B in our revised manuscript.

13. *Figure 7: My expectation would be that a final figure tries to summarize the key findings of a study. Since this is barely the case here, thus please consider to move it to the Supplement. Please check also the grammar of the legend (“” ... Of The Nanog regulation... ”).*

Answer: Thank you for the suggestion. We move the figure to supplementary figure 5 with the grammar correction of the legend of the revised manuscript.

REVIEWERS' COMMENTS:

Reviewer #1 (Remarks to the Author):

Thank you, the manuscript is significantly improved and my comments have been addressed

Reviewer #3 (Remarks to the Author):

The authors addressed properly all my points, and overall the manuscript improved substantially. I also like very much the Supplementary Methods which will be of great help for readers interested in testing this cultivation method.

There is only the following minor points that need some attention:

- Legend for Figure 2 starts with "Figure 1"
- Please revise grammar of this sentence (line 154): "Interestingly, when the cell viability was significantly reduced altogether with increasing lactate concentration, the glucose was still can be detected in suspension culture without any medium replacement in C/MR(-).

Answer to the Reviewer # 1

Thank you, the manuscript is significantly improved and my comments have been addressed.

Answer : Thank you very much for your careful reviewing and giving the important suggestion which improved our manuscript.

Answers to the Reviewer # 3

The authors addressed properly all my points, and overall the manuscript improved substantially. I also like very much the Supplementary Methods which will be of great help for readers interested in testing this cultivation method.

Answer: Thank you very much for your careful reviewing and giving the important suggestion. Your important comment and suggestions was improved our manuscript.

Minor comments and suggestions:

1. *Legend for Figure 2 starts with “Figure 1”.*

Answer: We are sorry for the error. We replaced the legend for Figure 2 with its proper legend.

2. *Please revise grammar of this sentence (line 154): “Interestingly, when the cell viability was significantly reduced altogether with increasing lactate concentration, the glucose was still can be detected in suspension culture without any medium replacement in C/MR(-).*

Answer: We are sorry for the insufficient grammar. We improved the sentence as follows: “Interestingly, despite the increasing level of lactate concentration and depletion in cell viability, the glucose level was remained detected in the group without any medium replacement in C/MR(-).”